# Experiences of transgender persons in accessing routine healthcare services in India: Findings from a participatory qualitative study

**Harikeerthan Raghuram**[1,2]*, **Sana Parakh**[1,2], **Deepak Tugnawat**[1,2], **Satendra Singh**[1,3], **Aqsa Shaikh**[1,4], **Anant Bhan**[1,2]

1 Initiative for Health Equity Advocacy and Research (iHEAR), Bhopal, India, 2 Sangath, Bhopal, India, 3 Department of Physiology, University College of Medical Sciences, Delhi, India, 4 Department of Community Medicine, Hamdard Institute of Medical Sciences and Research, New Delhi, India

* harikeerthan.raghuram@sangath.in

**Data Availability Statement:** Relevant excerpts from data supporting the findings of this article are

## Abstract

Despite having a higher burden of health problems, transgender persons face challenges in accessing healthcare in India. Most studies on healthcare access of transgender persons in India focus only on HIV related care, mental healthcare, gender affirmative services or on the ethno-cultural communities or transgender women. This study fills this gap by focusing on diverse gender identities within the transgender community with a specific focus on experiences in accessing general or routine healthcare services. A qualitative descriptive approach was used in this study. 23 in-depth interviews and 6 focus group discussions were conducted virtually and in-person with a total of 63 transgender persons in different regions of India between May and September 2021. The study used a community-based participatory research approach and was informed by the intersectionality approach. Thematic analysis was conducted to analyze the data. Four key themes emerged: (i) intersectional challenges in accessing healthcare start outside of the health system, continue through cis-gender-binary-normative health systems that exclude transgender persons; and at the interface with individuals such as health professionals, support staff and bystanders; (ii) the experiences negatively impact transgender persons at an individual level; (iii) in response, transgender persons navigate these challenges across each of the levels: individual, health system level and from outside of the health system. This is a first of its kind qualitative participatory study focusing on routine healthcare services of transgender persons in India. The findings indicate the need to move conversations on trans-inclusion in healthcare from HIV and gender affirmative services to routine comprehensive healthcare services considering the higher burden of health problems in the community and the impact of poor access on their lives and well-being.

## Introduction

Discourses in healthcare are often limited to the binary of male and female, excluding transgender persons—whose gender identity is different from the sex assigned at birth—as well as

made available within the manuscript and uploaded as supplementary information.

**Funding:** This study was funded by the Thakur Family Foundation, Inc. in the form of a grant (GT378). The funders had no role in study design, data collection and analysis, decision to publish, or preparation of the manuscript.

**Competing interests:** I have read the journal's policy and the authors of this manuscript have the following competing interests: Aqsa Shaikh and Anant Bhan are currently serving as Section Editors on PLOS Global Public Health. This does not alter our adherence to PLOS ONE policies on sharing data and materials.

gender non-binary and gender non-conforming persons whose gender identity is outside of the typical binaries of male and female [1, 2]. In this paper we use the word transgender as a broad term inclusive of other gender non-conforming identities. Transgender persons have a higher burden of health problems but face increased barriers in accessing quality healthcare [3]. In India, 4.88 million (0.04%) persons chose the 'other' category for the question on sex in the 2011 census survey (there was no separate question on gender) [4, 5]. This number, however, is potentially a significant undercount of transgender persons as many choose to hide their transgender identity due to stigma or may have been counted within 'male' or 'female' as there was no separate question on gender in the census survey [5].

The negative experience faced by transgender person contrasts with the recognition and acceptance of trans and gender nonbinary identities in India such as in Hindu mythology, ancient Tamil literature and medieval Indian royal culture [6, 7]. For example, the character of Shikhandi in the Indian epic, Mahabharata is that of a transmasculine person [6]. Many in the *hijra* community in India, predominantly comprised of transgender women, trace their roots back to a blessing from Lord Rama in the Ramayana, because of which they continue to attend significant life events to give blessings in return for money and gifts [7]. Thus, India has a strong history of ethnocultural transgender identities but their employment has been limited to give blessings (*badhai*) or to go for begging or sex work until very recently [2].

Part of the reason for this was that it was not until 2019 that Indian law explicitly recognized and protected the rights of transgender persons, under The Transgender Persons (Protection of Rights) Act, 2019 following a verdict from the Supreme Court of India in 2014 [8]. This legislation was in response to a verdict from the Supreme Court of India in 2014 under the National Legal Services Authority v. Union of India [9]. Prior to 2014, in addition to not being given constitutional rights, the transgender community was also criminalized under versions of colonial legislations such as The Criminal Tribes Act [10].

With respect to healthcare, The Transgender Act, 2019 recognizes the lack of access to healthcare and exclusion in medical education and practices and calls for reform in these areas [8, 11]. This was required as transgender persons' access to healthcare is limited and this is made challenging by various reasons. Underlying economic exclusion, lack of employment opportunities, lower rates of education including medical education, poor housing, reduced access to gender-concordant legal documentation play a pivotal role in putting the community at a higher risk of health problems and reduces health seeking behavior [2, 12, 13]. Within healthcare settings, transgender persons experience stigma and discrimination from health providers and feel excluded in health facilities that are designed in the binary of male and female and are often neglected [14–16]. Such negative experiences with health providers, can adversely affect their mental health and wellbeing [17].

These challenges are exacerbated in the light of specific healthcare needs within the transgender community. For example, the transgender community is at the higher risk for HIV and other sexually transmitted infections, which is related to a higher prevalence of sex work within the community [2, 18]. Because of minority stress, i.e., the psychological impact of belonging to a minority community with these challenges, there is a higher burden of mental health problems including substance use [19–22]. Further, transgender persons are also at a higher risk of other non-communicable diseases such as diabetes, stroke, and cardiovascular diseases [23, 24] considering limited and challenging healthcare access. Finally, transgender people have specific healthcare requirements related to gender affirmative services such as hormone therapy, which is essential, life affirming and lifesaving [25–27]. For many of these needs there are specific provisions within the Transgender Act including a comprehensive health insurance scheme [8, 11]. Implementation, though, remains a gap [11].

Studies focused on the experiences of transgender persons in accessing general healthcare services are limited [5]. A scoping review from 2020 shows that most studies in India focus on HIV related care, mental healthcare, gender affirmative services or focus on the ethno-cultural communities or only transgender women [5]. This study, titled TransCare COVID19, fills this gap by focusing on diverse gender identities within the transgender community using a qualitative participatory approach. The study examined various aspects such as access to healthcare, mental health care and impact of COVID19. This paper focusses on the experiences in accessing general or routine healthcare services. The paper also explores the impact of these experiences and how communities navigate their healthcare access.

## Methods

### Study design

The study used a community based participatory approach and was guided by the theoretical approach of intersectionality [28, 29]. The details of how these approaches were instrumentalized are described in another paper from the same study and hence is briefly mentioned in this paper [30]. For the community based participatory approach, the study included (i) a co-investigator and team members from the transgender community, (ii) a series of community workshops aimed at finalizing the research questions, the interview guide, sample, reviewing the data analysis and planning the next steps in light of the findings; (iii) recruitment in partnership with community based organisations; (iv) an advisory board including transgender community members; and (v) co-collection of data along with transgender community members.

An intersectionality approach was chosen based on prior research with the community by one of the study team members which suggested diversity of experiences within the community because of intersectional identities linked to language, education, class, caste, etc. [31]. Intersectionality approach seeks to examine the unique experiences individuals face because of different intersecting identities, some that give privilege, and others marginalizing, in reference to the context [28, 29]. In this study intersectionality approach informed the interview guide, the sampling and the data analysis.

Data was collected from May 2021 to October 2021 using the methods of In-Depth Interviews (IDIs) and Focus-Group Discussions (FGDs). IDIs have the potential of giving broad and detailed information from a few sources along with nuance and depth. FGDs were included into the study based on inputs received from the community as well as to capture intra-group differences and other dynamics.

The definition of transgender persons included persons belonging to ethno-cultural transgender identities such as *hijras* and *kinnars* in the sample as well as gender non-binary, gender non-conforming and gender fluid persons. *Hijras* and *Kinnars* constitute a varied community primarily consisting of individuals assigned male at birth. They hold non-conforming gender identities, leaning towards femininity in their expression, and often possessing an attraction to males. They primarily dwell within kinship-based groups known as *gharanas* or *dera* [32, 33].

The study was pan-India and not regionally focused. Purposive sampling was used with an aim to get in-depth and comprehensive information about the research question with a diversity of participants in line with the intersectionality approach. For example, the sampling included participants of different class, caste, region, cultural identity, gender identity and sexual orientation.

A total of 23 IDIs and 6 FGDs were conducted to achieve saturation and ensure maximum possible intersectionality that can be accommodated within the project constraints. Three participants took part both in an IDI and in an FGD. Thematic saturation was determined by

**Table 1. Sample characteristics.**

| Interview type | Participants | Number |
| --- | --- | --- |
| **In-depth interviews** | Hijra/Kinnar community | 3 |
| | Transmasculine persons | 6 |
| | Transfeminine persons | 12 |
| | Gender fluid/ non-binary persons | 2 |
| | Total | 23 |
| **Focus group discussions** | Hijra/Kinnar community | 22 |
| | Transfeminine persons | 11 |
| | Transmasculine persons | 5 |
| | Gender fluid/ nonbinary persons | 5 |
| | Total | 43 |

ongoing analysis along with team discussions and community workshops. Details of participants is given in Table 1.

## Data collection

WhatsApp outreach and phone calls were used for recruitment along with a social media post on Instagram with a Google form for potential participants to express their interest. In the social media post due measures to maintain confidentiality were taken such as turning off comments. Rapport building was easier due to existing trust in the connecting individual and in co-investigator AS (Aqsa Shaikh) who acted as a catalyst in the process. In addition, the investigator group engaged with specific community-based organizations of different intersections of identities to ensure diversity.

With participants fitting the inclusion criteria, the participant information sheet was shared and a pre-interview call was held to clarify any doubts and concerns to obtain a written informed consent. They were also made aware of their rights as research participants and the participant requests were accommodated to the best of our capabilities. The participants were given the option to have a person from the community on-board during the data collection if that felt safer. Translation facilities were also made available. Prior to initiating data collection, the researchers were trained in gender sensitivity with a focus on interviewing techniques by AS.

The IDI and FGD guide were semi-structured and developed in consultation with community members. The guide had three parts: (i) on experiences in accessing healthcare; (ii) intersectionality; and (iii) impact of COVID19. A paper discussing findings from the third part of the guide is published separately [30]. This paper focusses on the first two sections of the guide. For these two sections, although the data was collected during the COVID19 pandemic period, participants were asked to primarily recount experiences unrelated or prior to the COVID19 pandemic.

Data was primarily collected virtually, mostly on Zoom, because of restrictions due to the COVID19 pandemic. A few IDIs were telephonic as the participants did not have access to Zoom or good internet connection. Three in-person FGDs were held with ethno-cultural communities in different cities in North and Central India. They were conducted in person as it was felt that it may not have been otherwise possible to reach this population. In the in-person data collection, all precautions under the COVID-19 protocol were taken. The IDIs and FGDs lasted an average of around 45 and 120 minutes each.

All the IDIs and FGDs were conducted by co-authors SP, HR, AB and AS. In all FGDs, a co-facilitator was present who was either a person from the same community or a person with

experience and familiarity with the specific community group. HR dropped out of interviews when it was felt that a cis-het male presence might be making the participant uncomfortable. Some IDIs had a person from community for creation of a safer space. One IDI included an intern with prior permission of the participant ensuring optimal comfort level.

The recordings of the IDIs and FGDs dependent on the mode of the data collection (e.g.: phone calls, zoom calls or in person). In Zoom, video recordings were removed and only audio files were retained and stored in Sangath secure cloud storage. During in-person data collection, audio recorders (Sony) were used for audio recording, and the recording were downloaded and stored securely in Sangath cloud storage. All audio recordings were transcribed in a uniform format which allowed for analysis of all the data collected–in-persons and virtual–together.

## Data analysis

An inductive thematic analysis was used to analyze the data [34]. Transcriptions were done by SP with support from a colleague from another team and consultants and who signed a confidentiality agreement. The transcripts followed a set format with time stamp, speaker name and text in a tabular form. The basic information regarding the interviewer, duration, type etc. were also included in the transcripts. All identifiers, including but not limited to name, place of residence, name of hospitals or healthcare professionals, were omitted from the transcripts to maintain anonymity. The participants were given an option to access their transcripts if they wished to.

After each interview emerging themes were discussed between SP and HR as well as with the other co-investigators in weekly team meetings. Coding was done using Microsoft Excel by SP and HR with support from project interns. SP reviewed transcripts coded by others to ensure that no themes were missed out. Initially, the first 14 transcripts were open coded to cover various overt and covert themes that emerged during the interviews. Then the codes that emerged were organized into a framework by SP and HR that was then used to deductively analyze the remaining transcripts. However, new codes that emerged were also added. The framework was iteratively developed through discussions in the team and with the community in a community workshop. Ethics approval for the study was received from the Sangath Institutional Review Board. Informed consent was taken from all the participants in the study, digitally, either in a written format or verbally and recorded.

## Results

The results from the study are described across four themes which are also summarized in Fig 1.

## 1. Experiencing in accessing healthcare

Transgender persons' experiences in accessing healthcare were very varied and intersectional. Challenges started before reaching the health system, through cis-bi-normative health systems and with individuals.

**1a. Experiences started before reaching the health system.**   Intersectional aspects of being a transgender person made it difficult to access healthcare even before reaching the health system. Many transgender people lost the support of their natal families. Where they did have support, it made a big difference in their access to care as well as in their general life choices. Within healthcare the use of correct pronouns by a parent made a positive difference.

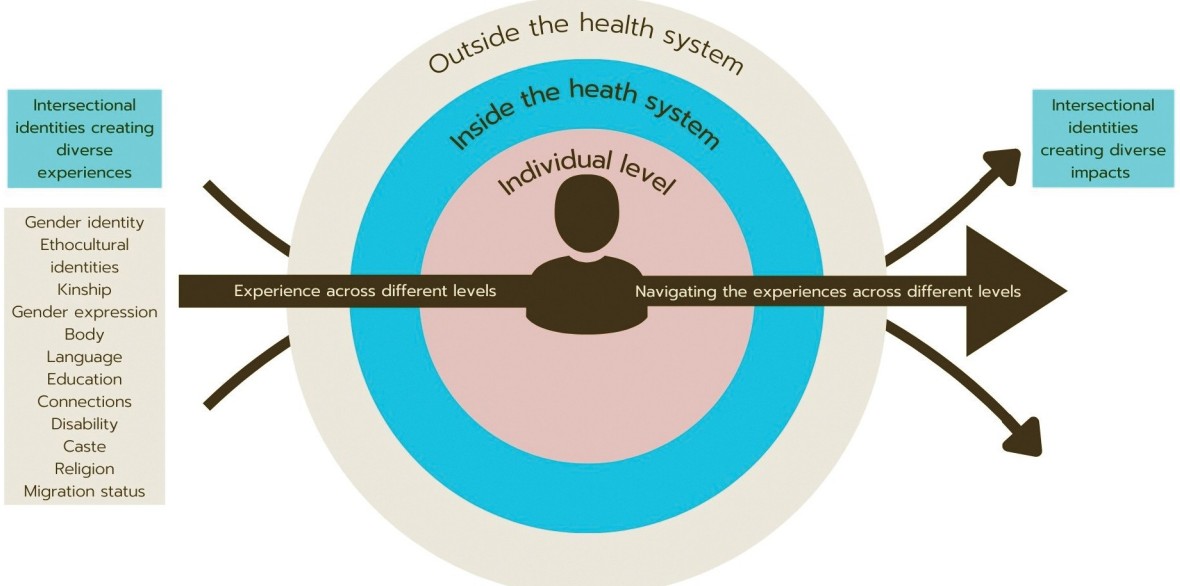

**Fig 1. A concept map summarizing the intersectional access to healthcare of transgender people.**

*I have never faced any sort of gender discrimination because my family has always supported me. There are many transgender people who don't get family support or are thrown out of their homes or are treated very badly–P43, A transwoman from Central India*

*She would use the pronoun "they" in front of the doctor. That really took some burden off my shoulders. –P16, A transmasculine person from South India*

For those who lived away from their natal families, friends, community-based organizations, and other kinship structures, such as the *dera* or *gharana* system, were their sources of support. For people living in the *dera* system, their access to care depended on the ability to get permission and support from the guru. Thus, the form of kinship support is a crucial aspect of each person's intersectional identity.

*Community support is something I really believed in since the time I was in college.–P23, A gender fluid person from Northeast India*

*If my friend wasn't there, I wouldn't have been able to access all these mental healthcare, trans healthcare options.–P20, A transmasculine person in an FGD*

*Yes, we get permission. They [Nayak/guru] never refuse and if there is any problem, they come along with us to the hospital, also.–P27, A kinnar person in an FGD*

Decisions on which doctor or healthcare center to visit were made very carefully because there was hesitation to try new doctors who were not known directly or indirectly unless it was an emergency. And for many it was based on referrals by others trans persons in their social networks.

*Other trans people said, "we go to him. And he's okay. He's fine. He's cool." And that's how I heard his name. . . I always prefer visiting [health professionals] on the recommendation of fellow trans people"–P49, A transwoman from Eastern India*

'Travelling to the health center was also a challenge because of experiences of stigma en-route to the hospital. This stigma was felt not only by verbal remarks made by other people but also by the general body language of people that reflected their attitude. For example, many participants spoke about the way people look, stare, their 'gaze', and laugh, that made them feel uncomfortable. This was also determined by how 'visibly trans' someone is (explained later).

*If you walk on the road, people will identify you as a trans person. . . a lot of people have myths and misconception. Either they are afraid of transgender, or they have pity, or sympathy.–P66, A transwoman from North India*

**1b. Experiences in cis-binormative health systems.** Within the healthcare system there were several positive and negative experiences. Almost all the participants we interviewed had at least one negative experience to share. Some of this was related to the health system, while others were with individual healthcare providers.

Many hospitals require service users to provide an identity card for registration. Despite recent legislative changes and numerous initiatives, transgender persons spoke of challenges in getting legal documents that match their gender identity. Lack of gender concordant identity documents resulted in problems at the registration. Gender discordant ID cards forced participants to share their trans identity to health professionals, some of whom were insensitive about it, for example, by laughing. Lack of ID cards sometimes led to denial of care. One participant shared about how she was asked for an ID proof only after the healthcare worker realized that she is a trans person.

*I was asked for an ID proof. I said, "why do you need ID proof to provide a medical treatment?" . . . After arguing for 10–15 minutes the doctor plainly denied seeing me.–P18, a transwoman*

*When the [Nurse or Doctor] used to see that our identity proof is on a male name. . . they used to giggle among themselves. There used to be a lot of misgendering–P16, A transmasculine person from South India*

The way hospitals and clinics are structured is often exclusionary to transgender people. Participants shared their feelings that the hospitals "don't expect a trans person" to come. This is because most hospitals segregate most spaces based on the gender-binary of male and female, thus excluding people who are non-binary or don't appear to fit within the norms of the gender expression in the binary. The spaces inside the hospitals include wards, outpatient clinics, waiting rooms, washrooms, and queues in registration. When transgender people arrive at such binary spaces there is confusion about which side to go to because of chances of being told they are in the 'wrong place' even when they identify in the gender binary. Registration forms are also usually limited to male and female without a separate transgender, non-binary or other column. Furthermore, specific specialties such as gynecology is normatively designated for cisgender female patients only. One participant spoke of hesitating to go to hospitals because of such issues.

*If I go into the men's washroom, I can't go with a saree. If I go into women's washroom, my voice gives it away–P40, a transwoman from North India*

"One transman spoke about how a gynecologist joked that once he transitions completely, he won't be allowed to the gynecology clinic which made him feel uncomfortable and unwelcome. This shows the deeply gendered norms within specific medical specialties.

I also informed [the gynecologist] that I am going to be transitioning and taking the injections. She laughed and made a joke about it and said, "Oh, so the next time you come here, you won't be allowed, because then you will look like a man." That just made me feel uncomfortable and unwelcome there.–P16, A transmasculine person from South India Some participants questioned this cis-bi-normative model and suggested alternatives such as asking the patient which ward they would be comfortable in, where it is binarily gender segregated. There have been few examples of hospitals and clinics where segregation based on gender is avoided, or there are separate queues for transgender persons or are exclusively for transgender persons. But in most places when a transgender person visits, there is confusion, neglect, and discrimination. The experience also depends on how 'visibly trans' someone is as seen earlier. Being able to pass as cisgender persons made it easier which is linked to one's gender expression, body and social norms. This is another intersection in identity that is crucial to defining a transgender persons experience in healthcare.

*In the in-take form itself hospital should ask, which wards they are comfortable in.–Transmasculine person in an FGD*

*I quite like the way that [name of a renowned institution] has structured the inpatient wards . . . there is no gender segregation of the washrooms–P64, a transwoman from South India*

*[Hospitals can have] segregated sections for every patient. [so that] every patient is guaranteed their privacy—whether it be using curtains or makeshift walls.–Transmasculine person in an FGD*

However, one transwoman spoke about how her ability to pass as a cis woman made her more anxious as health professionals never realize she is a transwoman with unique health needs. Therefore, she felt it might be easier for her if she was not so cis-passing and looked more gender fluid or non-conforming.

*If I was less passable as a woman and appeared more as gender fluid or as non-conforming, perhaps the level of anguish I would have gone through would have been lesser.–P1, a transwoman from Western India*

**1c. Experiences with individual practitioners, support staff and bystanders.** After crossing the hurdles in how hospitals are designed, then experiences with individual practitioners are a major challenge. This included neglect. While some trans participants were neglected by denial of care even before getting to see the health professional, another transwoman activist was asked to give sponge bath to a trans patient admitted because the nurse said that they are not comfortable "doing such things". This indicates neglect and exclusion of trans persons by health professionals because of stigma.

*The watchman never used to allow us to sit also. And they shoo away the people like that.– P66, a transwoman from North India*

*The nurse will not even touch your body because they feel it is obnoxious to touch a body of a trans person. So, they used to call me for some patient saying that "please come and give her the sponge bath" and I usually say "why can't you give it? You are a nurse you should be giving". Then they say "No, no, can you do it please? You will feel more comfortable doing it. People like us don't feel so comfortable doing such things"–P66, a transwoman from North India*

Consent while doing invasive procedures, such as per-vaginal examination, and privacy was a challenge for many participants. Many trans women spoke of experiences of being asked to undress without the presence of female doctors or nurses. A transman in an FGD spoke of how he felt uncomfortable undressing in front of male nurses but was forced to do it. This suggests the need for understanding of different trans identities and the need for sensitivity when during invasive examinations.

*[When asking to undress] the [male nurses] would shout saying "You are a boy, aren't you? Then why are you shy?" So, that was very uncomfortable for me. So, I didn't do it. But they kept pushing saying "uthao, uthao, uthao. . . [lift, lift, lift]". And there was no woman around. I was not very comfortable with it, but I had to do it.–Transmasculine person in an FGD*

*I was asked to undress in front of a male doctor. And it was very traumatizing experience for me. I kept asking for a female doctor. But they did not give me–P66, a transwoman from North India*

Participants spoke of different forms of abuse: financial, verbal, and sexual. Trans persons were financially abused by overcharging. Another participant was denied care after being charged an unusually high consultation fee.

*When the trans person comes to the hospital setting, there is always a fleecing of money. Like, immediately, another 500 rupees will be added"–P40, a transwoman from North India*

Verbal abuse occurred during registration and other points of waiting. Sexual abuse was through inappropriate advances, remarks and touching. One participant spoke about how this made her feel that she was an 'untouchable' which is a word referred to Dalit persons who are an oppressed caste group in India. Transgender persons felt discriminated from cisgender patients.

*There was an argument between the nurse who [said] "aap logon ka bhi insurance hota hai— kya chutiyapanti hai?" [Do people like you also get insurance—What atrocity is this?]–P12, transwoman, North India*

*There have been instances when I was physically abused, as well. There have been instances like people staring, passing weird comments, saying words [that are] derogatory, calling out names loudly, using physical [advances], or trying to take advantage. So, is it really like I'm an untouchable human being?–P18, a transwoman*

*There have been instances where I have been sexually given passes through winks, or through touches, which was inappropriate.–P12, transwoman, North India*

Experiences at the individual level were not just limited to doctors and nurses but all the staff in the hospital. In addition, there were experiences from other patients and bystanders, especially in government hospitals.

*We try to avoid public health systems because of the judgments that we face over there: how health care providers judge you, ask awkward questions right from the time when you enter the health system; including the watchmen, the nurse and ward boys.—P66, a transwoman from North India*

*While you are waiting in the line its quite an embarrassing situation where people are looking at you. They are passing comments, they are making gestures, which are not appreciable to*

*you. [There's] always this feeling that somebody is talking behind you, and you feel uncomfortable about it.–P66, a transwoman from North India*

Participants shared about feeling surprised on realizing that health professionals did not understand transgender identities and the medical transition process. For example, one doctor did not understand the difference between a vagina created surgically (neovagina) and a vagina by birth (P66). This showed that the medical curriculum does not cover transgender health topics sufficiently. A transmasculine person shared about how it is expected that health professionals don't know about transgender lives as society defines being cisgender and heterosexuality as the norm. A transwoman activist and public health professional spoke about how "it is not their fault as it is not there in the syllabus" (P12).

*We all have grown up in a cis normative society, where cis is the "normal", cisgenderism is normal, heterosexuality is the normal. So, people just don't know better. We need more accessible, inclusive curriculum system for our future healthcare professionals. The doctor doesn't know better because they didn't study those things when they were students. Transmasculine person in an FGD*

Since health professionals often did not understand what it means to be a transgender person, they tried to learn about transgender lives during their interaction with transgender patients which ended up dominating that interaction. Both of this resulted in communication that was often felt to be intrusive and insensitive by transgender patients. One transmasculine person shared about how they feel quite frustrated about having to explain their gender every time they share about their gender with a doctor.

*When people realize your identity is that of a transgender woman or that of a transgender man, there is a certain level of, there is a high level of inquisitiveness. Now, whenever there is inquisitiveness, it always borders on your right to privacy.–P1, a transwoman from Western India*

*I don't have to sit and explain to the doctor about what my gender is. . . So, every time I go to the doctor, it's almost like I am doing a gender-sexuality session. And I don't think that is something I want to deal with.–P50, a transmasculine person from North India*

In all these experiences, other identities of the participants made a difference in the kind of experiences they faced within the health system. On one hand, some identities helped in accessing better healthcare such as high income, being English-speaking, educated and having connections with doctors. On the other hand, other identities made the experience worse such as having a disability, being of lower caste, minority religion and of being a migrant.

*I can speak English and I am educated. . . [this] sort of makes it easier for me to navigate the power relationship between the doctor and patient–P15, A non-binary trans person from South India.*

*. . .My father is in defense. So, he had this respect that, "this is our officer"–Transmasculine person in an FGD.*

## 2. Experiences negatively impact individuals

The layered barriers and challenges in accessing and navigating health care negatively impacted transgender persons. Many trans persons often delayed and hesitated accessing care

until their health condition became worse because of misgendering and other uncomfortable experiences. Many resort to self-medication or alternate forms of treatments instead of going to a registered practitioner.

> *And I've many times I've avoided going to the emergency room just because I didn't want to face all this misgendering and discomfort. So, I had a ruptured cyst in my ovaries. And I probably should have gone to the emergency room to check that out. But I was like, "No, I don't I'll just manage it at home."–P16, A transmasculine person from South India*

> *Going to a doctor is never my first instinct. I will go to a doctor when stuff gets worst. . . I try to self-medicate or do home remedies.–P53, A transman from Western India*

The negative experiences also resulted in mental health impact such as anxiety before visiting the hospitals and fear and feeling of hopelessness when they fall ill. Misgendering, deadnaming and often being forced to visit hospitals in the sex assigned at birth rather than their own gender resulted in gender dysphoria.

> *I'm very anxious and dysphoric before I enter the healthcare setting and while I'm receiving the healthcare. After going and getting whatever that I need, I feel a little hopeless. And I start questioning myself, whether I'm being dramatic. Whether I'm expecting too much from people. I feel down and sad after going to any healthcare setting. –P16, A transmasculine person from South India*

### 3. Navigating experiences across different levels

**3.1. At the interpersonal level.** Transgender persons navigated these negative experiences by taking up measures that are also across different levels. At the individual level many of them would visit healthcare centers without disclosing their trans identity (cis passing). This would make the experience gender dysphoric but would help avoid the negative experiences they would have faced as a 'visibly trans' person. The choice also depends on the urgency of the healthcare need; with people choosing to cis pass if it's a more urgent healthcare need.

> *When I pass as cis-gendered, everyone is just like–'don't really care'. Everything goes on like for any other person. . . If it's an urgency, then I pass off as a cis identity. But if it is not a life-threatening issue, I usually present myself as a trans identity–P18, a transwoman.*

Others took an attitude of quite resignation and self-blame. Others avoid certain kind of health care facilities where the experiences may be worse. This response was also intersectionally determined. For example, trans persons who had digital access and networks spoke about how they feel more comfortable with accessing healthcare digitally and via telephone (P53). A trans person from ethnocultural community chose to endure the negative experiences.

> *Most of the doctors do argue with us. But we can't say anything to them.–P55, a kinnar person from Central India*

> *I completely avoid government healthcare systems now.–A transman in an FGD*

To decide between avoiding, postponing, and enduring, transgender persons balanced between the possibility of discrimination at the healthcare setting with the potential of irreversible harm due to a worsened health condition.

*I choose the possibility of discrimination over irreversible harm. I have to choose the lesser evil, although mentally it's difficult.–P49, a transwoman from Eastern India*

**3.2. At the health system level.** At the health system level, the measures included taking action to demand attention, for e.g., by clapping loudly or raising one's voice. The distinctive clapping of hands is a characteristic gesture of various ethnocultural transgender communities in India–often to announce presence [35]. In this context, the clapping was not just them speaking out about mistreatment, but an assertion and a demand as a *kinnar* person. Thus, the response to lack of access was also linked to the intersectional identity of belonging to an ethnocultural community and the need to bring attention to that identity to able recognition for health access.

*If they ask me to stop and get in line, all I do is say "I will clap loudly" and after that they don't stop me.–P5, a kinnar person in an FGD*

Others navigated by going with a friend or as a group. In an FGD with kinnar transwomen, participants spoke about how they merely decide to go to another health facility if they don't like the services.

*They are clueless and so we go as a group.–P2, a transwoman from South India*

*If we don't like the services received in a hospital, we go to another hospital.–a kinnar person in an FGD*

Few trans persons spoke about the importance of making efforts to educate the health professionals, especially participants who were educated themselves. Some participants shared about how doing this is quite challenging as it takes time away from the health consultation.

*But when there is an occasion to be able to explain to them what being a trans person is and what they should be addressing me as that I make that effort because I think it's important to do that as well to bring about change.–P36, a transmasculine person from South India*

*If my community is misgendered, my community is the first person to lift their saris and to clap and abuse them rather than educating them.–P12, transwoman from North India*

*When you try to tell them "Don't use that name", or "don't address me with those pronouns" instead of having the conversation about your illness, it turns into a moral policing conversation.–A transmasculine person in an FGD*

**3.3. Outside of the health system.** Outside of the health system, transgender persons, especially those who were activists themselves, resorted to advocacy using the Right to Information Act, 2005 (RTI), through the courts as well as by meeting specific senior health officers and policy makers. Under RTI Indian citizens can seek information from under the control of government authorities [36].

*I'll file an RTI somehow. . . And with two, three. . . we build pressure.–P2, a transwoman from South India who is an activist*

*So, we had to file a case against them–P2, a transwoman from South India*

*So, [a transgender person] had to do a lot of advocacies to try to meet multiple people, senior hospital staff, all the way to the health minister. . . after a lot of struggles one bed was allotted in a hospital for trans patients.–P36, a transmasculine person from South India*

## Discussion

This was one of the first of its kind and largest qualitative studies looking at challenges in routine healthcare access among diverse transgender persons in India using a community based participatory research approach. The study showed that transgender persons had diverse challenges in accessing healthcare that started outside of health systems, were present through cis-binormative health systems, at an interpersonal level with different health professionals. The same was navigated at the interpersonal level, at the health system level and outside the health system. While the experiences were modified by different intersectional identities, the impact of these experiences necessitate reform to improve access.

A 2007 study from the UK showed that 29% of transgender persons felt that their trans identity resulted in negative experiences while accessing general healthcare services [37]. Often the challenges in accessing healthcare among transgender persons is primarily looked at in terms of the challenges in interactions with health professionals [5]. This study showed that negative experiences at the individual level were not just with health professionals but also with support staff and other bystanders [3].This shows the immense needs for sensitization of all staff in a healthcare setting in providing trans-affirmative healthcare which could be pre-service and in-service and continuum of care.

Current medical education is very cis-bi-heteronormative and excludes trans experiences [11]. Results showed that the resultant lack of knowledge often results in intrusive communication within the clinical encounter. Over the last few years several developments have occurred in the context of medical education in India towards making it inclusive of transgender persons [38]. Though the national regulatory body in India, the National Medical Commission, has taken the first step towards more LGBTQIA+-inclusive medical education by revising the curriculum for Psychiatry and Forensic Medicine and by releasing a directive in this regard, there is still a long way to go [38, 39]. There is also the need to innovate methods to teach concepts such as dignity and empathy which cannot be taught through didactic lectures [38].

However, would it be enough to train health professionals when health systems are themselves exclusionary to transgender persons by being cis-bi-normative? This starts from the hospital gate where trans and gender non-normative individuals are "shooed away" by hospital security and at the registration desk where participants were required to present gender-concordant identity cards. Although The Transgender Persons (Protection of Rights) Act allow for self-identification to get an identity card as transgender, to get identification within the binary of male or female there is still a requirement to undergo gender affirmation surgeries in which there is gatekeeping [8, 11, 40, 41]. This has been partly modified in the linked Rules where there is still a requirement for "medical intervention" for identification within the gender binary [11].

The cis-bi-normativity of healthcare spaces is even more evident in queues, washrooms and wards which are very often divided in the gender binary of male and female, thus excluding trans and gender non-binary persons. Specific specialties such a gynecology is considered exclusive to (cis)women, excluding transmen who would also need gynecological services [42]. Hence, efforts need to be made to avoid segregation in the gender binary wherever it may not be necessary.

Alternatively, as mandated by the Transgender Act, separate healthcare centers may be created for transgender persons which could follow the well-functioning models such as those of Mitr Clinics, and certain medical colleges [8, 11]. A system of accreditation of those who are affirmative, and directory of such practitioners is also worth considering. However, it is to be noted that such segregation of transgender persons to health centers that are exclusively for them risks "ghettoization" and furthering of stigma. Hence, in the long run a systemic reform

to make all health centers accessible to, and inclusive of, transgender persons is vital [43]. Nevertheless, in the short term when mainstream public health facilities are trans-exclusionary, such exclusive centers become necessary and are often the only way transgender persons can access dignified care.

Good healthcare necessitates a good understanding of the social context of transgender persons. For example, it is important to recognize, accept and engage with the various intersectional identities of trans persons such as their unique kinship structures [33, 44]. In consent seeking, when surrogates are required, health providers should not insist on biological families or marital partners and instead be open to chosen families and unmarried partners. The social context also needs to be kept in mind in post-hospital healthcare such as after surgeries and in chronic conditions. Laws also need to recognize non-biological kinship structures. For e.g., according to the Transgender Persons (Protection of Rights) Act, "family" means a group of people related by blood or marriage or by adoption made in accordance with law [8]. The Transgender Act should expand this to include the right to live "other transgender and gender-variant persons, in hijra gharanas, and other choice-based residences outside the legal structures of adoption, marriage, and birth-based kinship" [11].

Since social support, especially from biological families, makes a positive difference in the experiences of transgender persons, efforts need to be made for better support within biological families. Awareness and access to information may help biological families understand the specific needs of the individuals, the stigma they face and strengthens the support systems [45]. Relief measures for those who still get removed such as Garima Greh in India are also important when done in a sensitive and inclusive manner [46].

But would reform of health system be enough when wider social spaces are still trans-excluding and transphobic? For example, stigma in public spaces, especially transportation can prevent a transgender person to even consider travelling to a healthcare center [47]. Studies from the US and UK showed that 70–80% of transgender respondents experience harassment in public spaces [37, 48]. This shows the need for interventions outside of the health system to improve health access of transgender persons. Therefore, health departments need to work inter-sectorally and interdepartmentally rather than working in silos for a structural reform. When there are unaddressed factors outside of the health system, it would also mean that despite sincere efforts within the health system, health-seeking behavior from the community may still be seemingly limited [5, 47, 49].

Nevertheless, structural reform should not mean that a one-size fits all approach would work. There is much diversity within the transgender community and participants; intersectional identities changed their experience in the health system positively and negatively. The experience of a Hindi-speaking kinnar transwoman in Kanpur is rarely like that of an upper caste English-speaking transwoman in Bengaluru. Similar intersectionality of experiences has been found in the context of HIV care among transgender women in India [47]. Locations of gender, caste, class, education, region, orientation, disability create overlapping and unique barriers in access to healthcare and needs further enquiry [47, 50]. Hence, structural interventions to improve access to healthcare need to be intersectional and tailored to the different segments within the transgender community.

Finally, interventions to improve access to healthcare is important and urgent considering the impact of the challenges in access to care, such as hesitation to access care, self-medication and delaying of access until health conditions worsen [17, 51]. A 2021 qualitative study from India found delayed ART initiation and disengagement from care due to negative experiences with health providers [47]. Future studies need to investigate impact in terms of quality of life and health outcomes. Negative experiences also impact mental health often by resulting in gender dysphoria [17].

### Limitations of the study

The study was done during the COVID19 pandemic and hence most of the data was collected virtually. Hence, we were not able to interview many participants who did not have access to internet services. We tried to compensate this by conducting interviews and focus group discussions in-person following COVID-related restrictions in place. However, still most of our participants are from urban areas. Interviews with people from ethnocultural communities were limited to people from the hijra and kinnar communities. We were not able to interview people from other ethnocultural communities such as those in south India and northeast India. We were also not able to interview many participants from Adivasi communities (scheduled tribes), Dalit communities (scheduled castes) and minority religious communities such as Christian, Jain and Buddhist.

## Conclusion

This qualitative study examined experiences of transgender persons in seeking, accessing, and navigating routine healthcare services in India. Study results highlight that while negative experiences often experienced at the interpersonal level, are also often because of a cis-bi-normative health system. Further, health providers and the health systems inherently reflect the larger society that is cis-bi-normative and excludes transgender persons. Addressing these challenges require a comprehensive systemic and structural efforts that recognize the diversity within the community and acknowledge the intersectionality of their experiences. This can help make the health system responsive and inclusive of the health needs of transgender persons.

## Supporting information

**S1 File. Guide for in-depth interviews and focus group discussions.**
(DOCX)

**S1 Data. Supporting data–relevant excerpts.**
(DOCX)

## Acknowledgments

This paper is based on the TransCare COVID19 research project, which was funded by the Thakur Family Foundation, Inc.; GT378. The research team expresses gratitude to community members who participated in the diverse workshops centered on community-based participatory research, as well as the study advisory board. For transcription and translation tasks, recognition goes to Utkarsh B, Akanksha Negi, Varun Sharma, and Yogendra Sen. Study volunteer Varshini Odayar and intern Supraja C are acknowledged for their contribution. Deepak Sahu, Gadha Thachappilly, Manoj Rajkule, and Deepchandra are appreciated for their roles in coordinating and jointly conducting certain focus group discussions. Venu Pillai and Sanjana Singh are particularly appreciated for their support in recruitment and data collection. The manuscript underwent review and received valuable input from Sharin D'souza, for which the team is also thankful.

## Author Contributions

**Conceptualization:** Harikeerthan Raghuram, Deepak Tugnawat, Satendra Singh, Aqsa Shaikh, Anant Bhan.

**Data curation:** Sana Parakh.

**Formal analysis:** Harikeerthan Raghuram, Sana Parakh, Aqsa Shaikh, Anant Bhan.

**Funding acquisition:** Harikeerthan Raghuram, Deepak Tugnawat, Satendra Singh, Aqsa Shaikh, Anant Bhan.

**Investigation:** Harikeerthan Raghuram, Sana Parakh, Aqsa Shaikh, Anant Bhan.

**Methodology:** Harikeerthan Raghuram, Aqsa Shaikh, Anant Bhan.

**Project administration:** Harikeerthan Raghuram, Sana Parakh, Deepak Tugnawat.

**Supervision:** Deepak Tugnawat, Satendra Singh, Aqsa Shaikh, Anant Bhan.

**Validation:** Aqsa Shaikh.

**Visualization:** Harikeerthan Raghuram.

**Writing – original draft:** Harikeerthan Raghuram, Sana Parakh.

**Writing – review & editing:** Deepak Tugnawat, Satendra Singh, Aqsa Shaikh, Anant Bhan.

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
