## [Decision Letter · Decision Letter 0]

26 Oct 2023

PGPH-D-23-01919

Experiences of transgender persons in accessing routine healthcare services in India: findings from a participatory qualitative study

Dear authors,

Thank you for submitting your manuscript to PLOS Global Public Health. After careful consideration, we feel that it has merit but does not fully meet PLOS Global Public Health’s publication criteria as it currently stands. Therefore, we invite you to submit a revised version of the manuscript that addresses the points raised during the review process.

It is a very interesting topic, however, some items could be improved (see reviewers comments).

Some items should be more detailed and explained.

For instance, why the intersectional approach was chosen and what it is.

It is also recommended to explain how it is a participatory methodology. The only information is that you use interviews and focus groups. Is the manuscript part of a bigger project?

How impacted to data collection the merge of face-to-face interviews, zoom interviews, and telephone interviews?

Please submit your revised manuscript by 25th of November If you will need more time than this to complete your revisions, please reply to this message or contact the journal office at globalpubhealth@plos.org. Please include the following items when submitting your revised manuscript:

We look forward to receiving your revised manuscript.

Kind regards,

Maria del Mar Pastor Bravo, Ph.D.

Academic Editor

Journal Requirements:

2. Please provide separate figure files in .tif or .eps format only and remove any figures embedded in your manuscript file. Please also ensure all files are under our size limit of 10MB.

3. In the online submission form, you indicated that "The datasets presented in this article are not readily available because they were collected from a marginalised community. Requests to access the datasets should be directed to the corresponding author". All PLOS journals now require all data underlying the findings described in their manuscript to be freely available to other researchers, either 1. In a public repository, 2. Within the manuscript itself, or 3. Uploaded as supplementary information.

Additional Editor Comments (if provided):

Dear author,

Congratulation on your work. It is a very interesting topic.

Some items should be more detailed and explained.

For instance, why the intersectional approach was chosen and what it is.

It is also recommended to explain how it is a participatory methodology. The only information is that you use interviews and focus groups. Is the manuscript part of a bigger project?

How impacted to data collection the merge of face-to-face interviews, zoom interviews, and telephone interviews?

Reviewers' comments:

Reviewer's Responses to Questions

**Comments to the Author**

1. Does this manuscript meet PLOS Global Public Health’s publication criteria? Is the manuscript technically sound, and do the data support the conclusions? The manuscript must describe methodologically and ethically rigorous research with conclusions that are appropriately drawn based on the data presented.

Reviewer #1: Yes

Reviewer #2: Yes

2. Has the statistical analysis been performed appropriately and rigorously?

Reviewer #1: N/A

Reviewer #2: N/A

3. Have the authors made all data underlying the findings in their manuscript fully available (please refer to the Data Availability Statement at the start of the manuscript PDF file)?

Reviewer #1: Yes

Reviewer #2: No

4. Is the manuscript presented in an intelligible fashion and written in standard English?

Reviewer #1: Yes

Reviewer #2: No

5. Review Comments to the Author

Reviewer #1: This is a well articulated article on a very timely issue. I suggest a minor revision on the following

1. Although published in other article from the same study, I suggest the authors to write more on intersectionality and community based participatory approach in the methods section, so that readers get adequate idea about these approaches from this article.

2. What is AS? please elaborate when first used

3. in page 10, line 200 in quote, give an explanation in parenthesis what was the 'they' in their language, I mean what exact word they used.

3. intersectionality is an important concept used in this paper. However, data and analysis on intersectionality in not adequate. I suggest to consider intersectionality in presenting and explaining each experiences in finding section 1, rather than making a small section of findings on 'intersectionality'.

4. this article is overloaded with quotes. However, overall the explanations of quotes should be more in-depth and detail. In these cases, the authors may like to remove some quotes and give more explanations and analysis of the quotes.

5. the authors can also draw a diagram/framework on challenges transpeople face in accessing heath care from an intersectional lens. This will help theorise the findings.

Reviewer #2: I would like to congratulate the authors on this study that fills an important gap in the literature. I strongly feel that the study makes significant contribution to the field by bringing alive the lived experiences of transgender community, and the direction it provides for the training of medical professionals in particular. I would like to make few minor but important points that I think would enhance the readability as swell as potential impact of the research work.

1. Authors should provide a brief narration with respect to the status the third gender and their rights in India and its potential implications. This background is important in light of the wide international readership of the journal.

2. Authors are requested to provide more details on the sampling. While authors mention that the participants are mostly urban, yet there seems to be heterogeneity in geographical and socio-economic background (including age groups) that would lead to varied experiences as in the last 5-6 years there has been few monumental developments in the country with respect to LGBTQIA+ community.

3. Authors make a very important point that multiple identities other than gender too shaped the experiences both positively and negatively. However, this seems inadequate based on evidence provided. Authors are requested to elaborate this section with more evidence.

6. PLOS authors have the option to publish the peer review history of their article (what does this mean?). If published, this will include your full peer review and any attached files.

**Do you want your identity to be public for this peer review?** For information about this choice, including consent withdrawal, please see our Privacy Policy.

Reviewer #1: **Yes: **Sanzida Akhter

Reviewer #2: No

---

## [Decision Letter · Decision Letter 1]

26 Jan 2024

Experiences of transgender persons in accessing routine healthcare services in India: findings from a participatory qualitative study

PGPH-D-23-01919R1

Dear Dr. Raghuram,

We are pleased to inform you that your manuscript 'Experiences of transgender persons in accessing routine healthcare services in India: findings from a participatory qualitative study' has been provisionally accepted for publication in PLOS Global Public Health.

Best regards,

Julia Robinson

Executive Editor

Reviewer Comments (if any, and for reference):

Reviewer's Responses to Questions

**Comments to the Author**

1. If the authors have adequately addressed your comments raised in a previous round of review and you feel that this manuscript is now acceptable for publication, you may indicate that here to bypass the “Comments to the Author” section, enter your conflict of interest statement in the “Confidential to Editor” section, and submit your "Accept" recommendation.

Reviewer #2: All comments have been addressed

2. Does this manuscript meet PLOS Global Public Health’s publication criteria? Is the manuscript technically sound, and do the data support the conclusions? The manuscript must describe methodologically and ethically rigorous research with conclusions that are appropriately drawn based on the data presented.

Reviewer #2: Yes

3. Has the statistical analysis been performed appropriately and rigorously?

Reviewer #2: N/A

4. Have the authors made all data underlying the findings in their manuscript fully available (please refer to the Data Availability Statement at the start of the manuscript PDF file)?

Reviewer #2: No

5. Is the manuscript presented in an intelligible fashion and written in standard English?

Reviewer #2: Yes

6. Review Comments to the Author

Reviewer #2: It was a pleasure reading your work and heartening to see an important issue like this being addressed through your research. I sincerely congratulate the authors for their valuable contribution to health sciences in this direction.

7. PLOS authors have the option to publish the peer review history of their article (what does this mean?). If published, this will include your full peer review and any attached files.

**Do you want your identity to be public for this peer review?** For information about this choice, including consent withdrawal, please see our Privacy Policy.

Reviewer #2: **Yes: **Sandip K. Agarwal, IISER Bhopal
